# Chemoprevention in BRCA1 mutation carriers (CIBRAC): protocol for an open allocation crossover feasibility trial assessing mechanisms of chemoprevention with goserelin and anastrozole versus tamoxifen and acceptability of treatment

Aideen M Campbell,[1] Melanie Morris,[2] Rebecca Gallagher,[2] Ruth Boyd,[2] Hazel Carson,[3] D Paul Harkin,[1] Ewa Wielogorska,[4] Christopher Elliott,[5] Kienan I Savage,[1] Stuart A McIntosh[1]

KIS and SAMI contributed equally.

For numbered affiliations see end of article.

**Correspondence to**
Stuart A McIntosh;
s.mcintosh@qub.ac.uk

## ABSTRACT

**Introduction** BRCA1 mutation carriers have a significant lifetime risk of breast cancer, with their primary risk-reduction option being bilateral mastectomy. Preclinical work from our laboratory demonstrated that in BRCA1-deficient breast cells, oestrogen and its metabolites are capable of driving DNA damage and subsequent genomic instability, which are well-defined early events in BRCA1-related cancers. Based on this, we hypothesise that a chemopreventive approach which reduces circulating oestrogen levels may reduce DNA damage and genomic instability, thereby providing an alternative to risk-reducing surgery.

**Methods and analysis** 12 premenopausal women with pathogenic BRCA1 mutations and no previous risk-reducing surgery will be recruited from family history clinics. Participants will be allocated 1:1 to two arms. All will undergo baseline breast biopsies, blood and urine sampling, and quality of life questionnaires. Group A will receive goserelin 3.6 mg/28 days by subcutaneous injection, plus oral anastrozole 1 mg/day, for 12 weeks. Group B will receive oral tamoxifen 20 mg/day for 12 weeks. Following treatment, both groups will provide repeat biopsies, blood and urine samples, and questionnaires. Following a 1-month washout period, the groups will cross over, group A receiving tamoxifen and group B goserelin and anastrozole for a further 12 weeks. After treatment, biopsies, blood and urine samples, and questionnaires will be repeated. DNA damage will be assessed in core biopsies, while blood and urine samples will be used to measure oestrogen metabolite and DNA adduct levels.

**Ethics and dissemination** This study has ethical approval from the Office for Research Ethics Committees Northern Ireland (16/NI/0055) and the Medicines and Healthcare products Regulatory Agency (MHRA) (reference: 32485/0032/001–0001). The investigational medicinal products used in this trial are licensed and in common use,

## Strengths and limitations of this study

► Solid underpinning preclinical data generated within our laboratory.
► Extensive patient and public involvement in study design and development.
► Oestrogen and oestrogen metabolite levels assessed using established, highly sensitive mass spectrometry-based methodology.
► Lack of qualitative intervention to support recruitment and investigate reasons for declining study participation.
► DNA damage is not currently a validated biomarker for cancer risk.

with well-documented safety information. Dissemination of results will be via high-impact journals and relevant national/international conferences. A copy of the results will be offered to the participants and be made available to patient support groups.

**Trial registration number** EudraCT: 2016-001087-11; Pre-results.

## INTRODUCTION

Women with a germline BRCA1 mutation have up to an 85% lifetime risk of developing breast cancer by age 70, with the majority of these women developing triple negative disease.[1 2] Intriguingly, a number of retrospective studies, including a large meta-analysis, have demonstrated that risk-reducing oophorectomy significantly reduces the risk of developing breast cancer in this population by up to 50%.[3 4]

In contrast, more recent studies, including a prospective study, contradict this, suggesting some protective effect in BRCA2 but not BRCA1 mutation carriers.[5 6] However, these studies have limited follow-up (mean follow-up period of 5.6 and 3.2 years respectively), and it is clear from chemoprevention studies carried out in large populations of women at increased risk that a limited chemopreventive effect is seen before 5 years, with significantly greater protective effects seen beyond 10 years.[7] Furthermore, oophorectomy in a mammary-specific BRCA1 knockout mouse model reduces mammary tumour formation as compared with non-oophorectomised mice.[8] Taking this together, the evidence suggests that oophorectomy may confer a protective effect in BRCA1 mutation carriers; however, long-term prospective data are required to support this. Currently, risk-reducing mastectomy is the mainstay of preventative treatment offered to BRCA1 mutation carriers, which is effective but carries significant associated costs in terms of physical and psychological morbidity and healthcare spending.[9–11] Aside from risk-reducing mastectomy, the only other risk-reduction strategy available to these women is chemoprevention.

In the UK, the National Institute for Health and Care Excellence recommends the selective oestrogen receptor modulators (SERMs) tamoxifen and raloxifene for use in women at high risk of developing breast cancer, on the basis that they have been demonstrated to reduce the incidence of oestrogen-receptor positive tumours.[12] However, evidence on the benefit of tamoxifen as a chemopreventive agent in BRCA1 mutation carriers is conflicting. One small study showed no benefit from tamoxifen in reducing breast cancer risk in BRCA1 mutation carriers, although it has been suggested that tamoxifen use in BRCA1 mutation carriers with breast cancer may reduce the incidence of contralateral tumours.[13 14] In one case–control study, short-term (2-year) tamoxifen use was as protective for contralateral breast cancer as a longer (5-year) course in BRCA1 and BRCA2 mutation carriers, suggesting some potential chemopreventive efficacy.[15]

However, given that the majority of tumours that develop in BRCA1 mutation carriers are oestrogen receptor (ER)-negative, it seems counterintuitive that SERMs will reduce breast cancer risk, given that no benefit in ER-negative tumours has been shown in the chemoprevention setting with tamoxifen.[12]

Clinical trial data from the Second International Breast Cancer Intervention Study (IBIS-II) suggest a reduction in breast cancer risk using aromatase inhibitors (AI) in women with increased risk.[16] However, there is no direct evidence to support a benefit for this approach in BRCA1 mutation carriers.

Clearly oestrogen plays a key role in tumourigenesis in these patients, as tumours develop preferentially in the oestrogen-rich tissues of the breast and ovary.

Preclinical data from our laboratory demonstrate that oestrogen and oestrogen metabolites cause DNA double strand breaks (DSBs) in ER-negative breast cells (normal and cancer cell lines, as well as primary breast progenitor cells isolated from patients), and that BRCA1 is necessary for repression and repair of these DSBs.[17] The implication of these findings is that in BRCA1-deficient breast cells, exposure to oestrogen and its metabolites is capable of driving genomic instability, a well-defined early event in BRCA1-related cancer development.[18 19]

Taken with evidence from clinical trials, which demonstrate no benefit from SERMs in reducing the incidence of ER-negative tumours, this suggests that SERMs are unlikely to be effective chemopreventive agents in BRCA1 mutation carriers (who are more likely to develop ER-negative tumours). Our data suggest that a chemopreventive approach which lowers circulating oestrogen levels and reduces exposure to oestrogen metabolites within the breast tissue may be more successful by removing this driver of genomic instability. While DNA damage is not at present a validated biomarker of cancer risk in mutation carriers, as described earlier, genomic instability is a well-characterised early hallmark of BRCA1-associated breast tumours. Consistent with this, mice harbouring specific point mutations within BRCA1 affecting its DNA repair function while conserving other protein functions (E3 ligase, transcriptional activity and so on) develop tumours.[20] This supports the link between DNA damage, genomic instability and tumourigenesis in BRCA1-deficient breast cells.

In postmenopausal women (or premenopausal women who have undergone oophorectomy), the use of an AI may be considered. In premenopausal women with intact ovaries, the induction of reversible ovarian suppression using a luteinising hormone releasing hormone agonist (LHRHa) in combination with an AI, to suppress subsequent aromatase-mediated oestrogen production, may be effective.

Progesterone signalling has also recently been implicated in BRCA1-related tumourigenesis, with the tumour necrosis factor superfamily member RANK (receptor activator of nuclear factor κ B) and its corresponding ligand RANKL (ligand of receptor activator of nuclear factor κ B) functioning as key factors in this pathway and potential targets for chemoprevention.[21] Importantly, LHRHa suppression of the hypothalamic-pituitary axis will also result in decreased progesterone levels, which will consequently reduce progesterone-dependent RANK/RANKL signalling. Additionally, expression of the progesterone receptor (PR), a critical mediator of the progesterone/RANK/RANKL pathway, is itself driven by oestrogen signalling through the ER. Taken together, the suppression of oestrogen is likely to provide chemopreventive potential, similar to or surpassing that demonstrated through RANK/RANKL suppression.

Although LHRHa, AIs and SERMs are used in breast cancer and fertility treatments, there are no published data in the context of BRCA1-specific chemoprevention where the intention is to prevent ER-negative breast cancer.

It is known that compliance and acceptability of these agents in the adjuvant setting of breast cancer treatment are good, although with a high frequency of side effects.[22]

The compliance of women taking SERMs for chemoprevention is suboptimal, with approximately 40% of women unable to complete 5 years of therapy because of side effects.[23] Therefore, compliance with LHRHa in combination with an AI in the chemopreventive setting must be assessed and compared with SERM data as an important consideration in the feasibility of use in this context. In line with this, this study aims to inform future tolerability and compliance evaluation in a cohort who may require long durations of treatment and have a long life expectancy.

If the results from this study demonstrate that these treatments are a feasible and tolerable strategy, and that DNA damage is reduced through reduced levels of circulating oestrogen, this would provide data to support the proposed mechanism of oestrogen-driven tumourigenesis in BRCA1 mutation carriers. Furthermore, this may provide evidence to support further studies evaluating the use of this chemopreventive strategy in BRCA1 mutation carriers. Nevertheless, this would require a multicentre international collaborative study, which presents significant challenges. Undoubtedly the treatment regimen proposed within this study has a clear side-effect profile, the tolerability and acceptability of which will be addressed in this study. Currently, a number of groups are investigating inhibition of progesterone and/or RANK/RANKL signalling as an alternative chemopreventive strategy in BRCA1 carriers, which may present a more tolerable approach.[24] However, a recent study using a well-characterised mouse model of BRCA1-associated breast cancer suggests that oestrogen is the predominant driver of tumourigenesis rather than progesterone.[25] Given that RANK/RANKL signalling is progesterone-dependent, the efficacy of these progesterone and RANK/RANKL-targeted chemopreventive approaches remains unproven. In light of this, continued exploration of oestrogen suppression in this cohort is justified in order to maximise the repertoire of potential chemopreventive options in this area of unmet clinical need.

## METHODS/ANALYSIS
### Aims, design and setting of the study
The principal aim of this research is to establish if these treatments are tolerable and to evaluate patient compliance. The secondary aims include tolerability of treatments and quality of life (QOL).

As exploratory objectives, this research seeks to confirm the link between oestrogen exposure and DSBs and generate further insight into how oestrogen and its metabolites affect tumourigenic transformation, and to generate preliminary data necessary to support the development of a larger clinical study to evaluate chemoprevention in BRCA1 mutation carriers.

The study aims and objectives are summarised in table 1.

| Table 1 Trial objectives and endpoints | |
| --- | --- |
| Primary objective | Primary endpoint |
| To establish the acceptability of trial treatments as a chemopreventive strategy in BRCA1 mutation carriers and compliance. | Percentage of patients given patient information sheet who consent to trial entry. Measurement of compliance with treatment using patient medication cards. |
| Secondary objective | Secondary endpoint |
| To establish tolerability of trial treatments and procedures. | Measurement of patient quality of life using questionnaires at baseline and after completion of each treatment arm. Recording of adverse events that occur during treatment. |
| Exploratory objectives | Exploratory endpoints |
| To establish the potential of chemopreventive agents to reduce oestrogen-mediated DNA damage in breast tissue in BRCA1 mutation carriers. | Measurement of DNA damage in breast tissue using comet assays and immunohistochemical assays at baseline and after completion of each treatment arm. |
| To establish the effect of treatment on oestrogen metabolite levels in breast tissue, urine and serum. | Measurement of breast, blood and urinary oestrogen metabolite levels using ultra performance liquid chromatography tandem mass spectrometry at baseline and after completion of each treatment arm. |
| To explore the mechanism of oestrogen-mediated DNA damage in breast tissue in BRCA1 mutation carriers. | Analysis of breast tissue samples using alternative methods of DNA damage assessment. |
| To explore the relationship between oestrogen metabolite concentration in serum and urine and DNA damage in breast tissue in BRCA1 mutation carriers. | Analysis of breast tissue, urine and blood samples for proxies of oestrogen levels and DNA damage. |
| To explore potential biomarkers of chemoprevention efficacy. | Analysis of breast tissue, urine and blood samples for potential biomarkers of chemopreventive efficacy. |

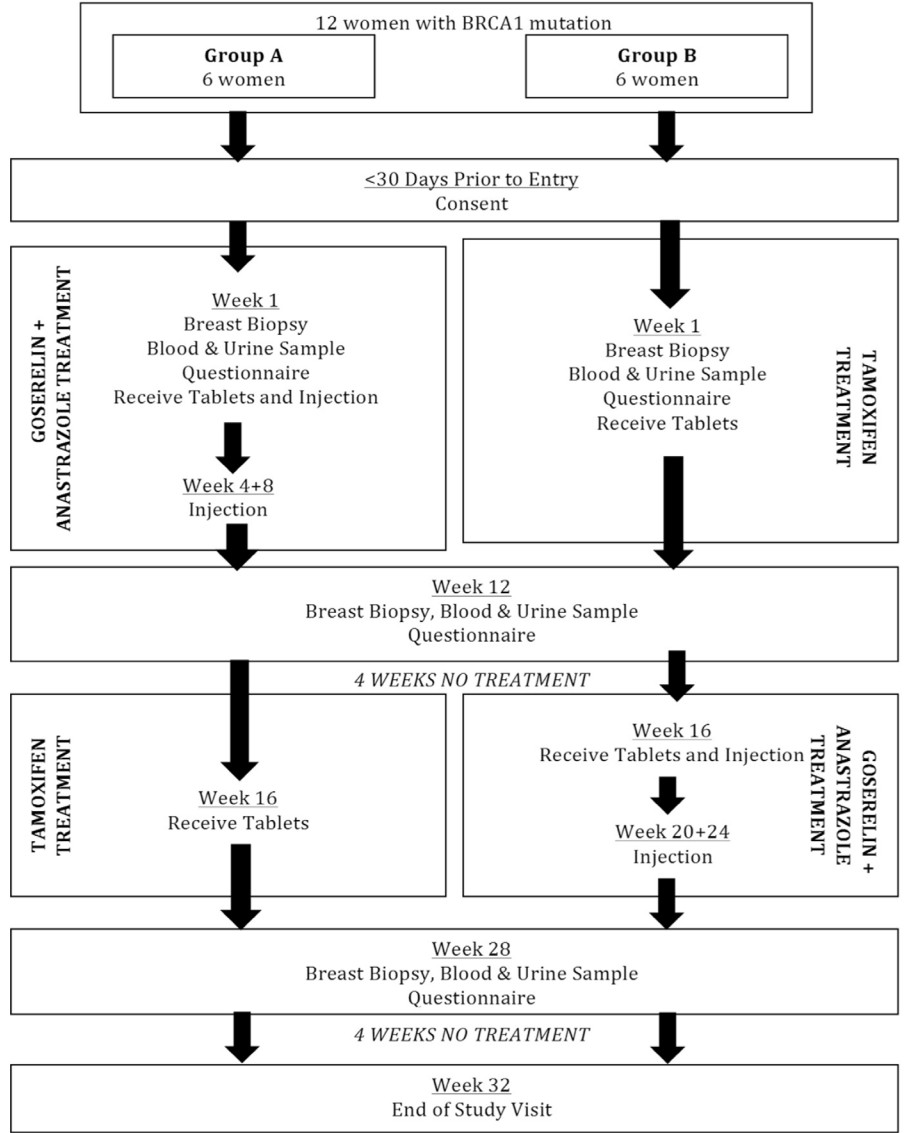

**Figure 1** Chemoprevention in BRCA1 mutation carriers (CIBRAC) trial schema.

## Design

A schematic overview of the trial is depicted in figure 1.

The crossover design allows comparison of outcomes from both treatments with the participant baseline; thus, each participant acts as their own control for both treatment arms. Blinding or masking is not achievable given the nature of administration of goserelin (subcutaneous implant). Given that each participant will receive both treatments without blinding, randomisation was felt unnecessary, and alternate allocation of treatment groups in order of recruitment is adequate. The setting of the trial in tertiary care (family history breast clinics) reflects the main location for the ongoing care and management of this cohort.

## Selection of participants

Participants will be recruited from family history breast clinics within Northern Ireland as identified by clinicians and by posters and publicity regarding the trial presented and distributed at relevant public engagement events. The participant inclusion and exclusion criteria are outlined in table 2. The chief investigator (or appropriately trained delegate) will confirm eligibility against the inclusion/exclusion criteria.

## Description of interventions
### Goserelin and anastrozole treatment

Participants will receive goserelin 3.6 mg by subcutaneous injection every 28 days for 12 weeks, simultaneously receiving oral anastrozole 1 mg per day for 12 weeks.

### Tamoxifen treatment

Participants will receive oral tamoxifen 20 mg per day for 12 weeks.

### Washout period

On completion of the first treatment arm, participants will have a 4-week period of no treatment to allow washout of active substances, before commencing second treatment arm.

**Table 2** Inclusion and exclusion criteria

| Inclusion criteria | Exclusion criteria |
| --- | --- |
| Age >18 years. | BRCA1 mutation of uncertain significance. |
| Premenopausal. | Personal history of breast or ovarian carcinoma. |
| Known pathogenic BRCA1 mutation. | Previous risk-reducing breast or ovarian surgery. |
| Intact ovaries. | Postmenopausal status. |
| No previous breast/ovarian carcinoma. | Concomitant use of alternative chemoprevention regimens. |
| No other previous malignancy. | Concomitant use of other hormonal agents less than 1 month prior to enrolment. |
| No previous use of chemoprevention. | Contraindications to study drug therapies. |
| Willingness to use non-hormonal contraception. | Contraindications to breast core biopsies. |
| | Pregnancy or breast feeding. |
| | Inability to give informed consent. |
| | Having made a decision to proceed with risk-reducing surgery. |

## Description of processes

### Procedure for obtaining informed consent and trial arm allocation

Participants will receive an adequate explanation of the objectives, methods, potential risks and anticipated benefits of the study. They will be given a current, ethically approved participant information sheet for their consideration. Following this, informed consent will be obtained once women have had a minimum of 24 hours to consider trial entry and the opportunity to ask further questions. Following consent, participants will be provided with a copy of the signed consent form, with a copy filed in the patient notes, and the original retained in the site file, available for inspection if required.

No protocol-required assessments will be conducted until the consent form has been signed and dated by both the participant and the investigator. Participants will be provided the option to allow the use of blood samples, other body fluids and tissues obtained during testing, operative procedures, or other standard medical practices for further research purposes. Participants will be alternately allocated to treatment arms.

### Replacement of participants

In the case of a participant request for withdrawal or failure to collect biological samples, the participant will be withdrawn from the study and a new participant enrolled into the treatment arm the original participant has been retracted from.

### Percentage consenting to trial entry

Records will be kept of all the patient information sheets distributed to allow calculation of percentage of patients consenting to trial entry.

### Compliance measurement

Participants will receive a medication diary card for each oral treatment at the same time the tablets (anastrozole for arm A and tamoxifen for arm B) are issued to track compliance.

### Concomitant medication

Participants must be instructed not to take any medications (including herbal and over-the-counter remedies) during the study without prior consultation with the investigator. Any hormonal treatments (including all forms of hormonal contraception) are prohibited. Concomitant medications must be recorded in the patients' notes at the time of initial consultation and any amendments documented at subsequent visits.

### QOL assessment

Questionnaires will be completed on day 1 (baseline), day 84 (±3 days, following the end of the first arm) and day 196 (±3 days following the end of the second arm).

Two questionnaires will be used to assess health-related QOL:
1. European Organisation for Research and Treatment of Cancer Questionnaire to Quality of Life of Cancer patients, assessing QOL.
2. European Organisation for Research and Treatment of Cancer Questionnaire module to assess quality of life of patients with breast cancer, assessing treatment side effects and sexual dysfunction.

### Recording of adverse events

An adverse event (AE) is any untoward medical occurrence (including deterioration of a pre-existing medical condition) in a subject administered an investigational medicinal product (IMP). The event does not necessarily have a causal relationship with treatment or usage. An adverse reaction (AR) is an AE that is suspected as having a causal relationship to the IMP. Each AE and its causality, severity and expectedness are assessed by the chief investigator (or nominated representative) and recorded in the patient notes. All IMPs in this study are licensed with well-defined safety profiles. As such, we will only record data relating to ARs in the case report form (CRF). If the event is classified as a serious adverse event (SAE), the reporting procedures as per Belfast Health and Social Care Trust (BHSCT) will also be followed. Follow-up will continue until all the necessary safety data for the event have been gathered and until the AR or SAE has resolved, returned to baseline or stabilised.

### Pregnancy testing

Participants will be asked to provide a non-sterile urine sample (the initial stream) in a clean, non-sterile receiver. The sample will be used to perform a urinary human

chorionic gonadotropin (hCG) test in the clinic following the manufacturer's standard operating procedure.

Pregnancy testing will be performed twice:

1. At the time of assessment of eligibility for the trial following consent, prior to the first arm treatment commencement.
2. On day 112±3 (week 16), prior to starting the second treatment arm.

### Core biopsies

Biopsy samples will be collected as standard using a 14G core biopsy needle under ultrasound guidance in the Department of Radiology at Belfast City Hospital. Two core biopsies will be sharply dissected into at least four smaller samples (approximately $5\,mm^3$ each) and placed in individual Eppendorf tubes, then snap-frozen with liquid nitrogen and stored for later use. Two core biopsies will be placed in individual formalin-containing sample pots, after which they will be formalin-fixed paraffin-embedded (FFPE) as per standard operating procedures within the Belfast Trust.

Core biopsies will be taken on day 1 (baseline), day 84 (±3 days following the end of the first arm) and day 196 (±3 days following the end of the second arm).

### Histopathological analysis

One FFPE sample will be assessed by a histopathologist to ensure adequate sampling of glandular tissue.

### Comet assays

Two frozen biopsy specimens will be used to perform tissue-based comet assays to assess DNA damage.

### Immunohistochemistry

One FFPE core biopsy sample will be sectioned and stained with antibodies against 53BP1 and phosphorylated histone A2 variant X (γH2AX) using conventional and fluorescent immunohistochemistry to assess DNA DSBs in these samples.

Additional sections will also be stained for RANK and RANKL as well as Ki67 using immunohistochemistry.

### Mass spectrometry

Two frozen biopsy specimens will be used to examine concentrations of oestrogen metabolites 2-hydroxyestradiol ($2$-$OHE_2$) and 4-hydroxyestradiol ($4$-$OHE_2$), using ultra performance liquid chromatography tandem mass spectrometry (UPLC-MS/MS).

### Serum samples

Serum sampling will be performed on day 1 (baseline), day 84 (±3 days following the end of the first arm) and day 196 (± days following, end of the second arm). Participants will undergo venepuncture with blood samples collected in two silica (clot activator) coated 10 mL blood sample tubes. After the minimum recommended clotting time (60 min), blood samples will be centrifuged to allow separation of serum from blood cells, with the serum stored at −80°C in 1 mL aliquots in RNAse free

tubes. Following sample accrual, samples will be thawed and serum $2$-$OHE_2$ and $4$-$OHE_2$ levels, and levels of depurinating DNA adducts (common types of DNA damage caused by oestrogen metabolites) will be measured using UPLC-MS/MS.

### Urine samples

Urine sampling will be performed on day 1 (baseline), day 84 (±3 days following, end of the first arm) and day 196 (±3 days following, end of the second arm). Mid-stream urine samples will be collected in sterile containers. Within 2 hours of collection samples will be centrifuged to separate debris from the urine and then stored at −80°C in 500 μL aliquots in RNAse free tubes. Following sample accrual. Samples will be thawed and urinary $2$-$OHE_2$ and $4$-$OHE_2$ levels and levels of depurinating DNA adducts will be measured using UPLC-MS/MS.

### Whole blood sampling

Each participant will provide one whole blood sample at one time point in the study. This will be performed at baseline (day 1), unless the patient is already enrolled and receiving treatment, in which case the whole blood sampling will be performed at the end of the treatment arm that they are currently receiving (day 84±3 days or day 196±3 days).

The whole blood samples will be collected at the same venepuncture session as serum sampling. Blood will be collected into one EDTA-coated tube and frozen/stored at −80°C in 1 mL aliquots. Samples will be retained for future use including DNA extraction.

The timeframe for interventions and assessments is summarised in table 3.

### Comparisons

Each participant acts as her own control. The results from each individual's set of samples (core biopsies, serum and urine) and questionnaires, taken at baseline, end of the first arm and end of the second arm will be compared. The results will not be compared between patients.

### Statistical analysis

This study is primarily aimed to determine acceptability and tolerability of the proposed chemopreventive treatments. Therefore, no formal sample size calculation has been carried out, as the laboratory endpoints are all exploratory.

For these reasons, no formal statistical analysis is planned. Nonetheless, descriptive statistics will be used to summarise the principal and experimental findings of the study.

### Laboratory analysis

The proposed translational studies include performing comet assays on fresh-frozen breast core biopsies at baseline and after each treatment arm. The comet assay protocol has been optimised on breast cell cultures as well as fresh-frozen breast biopsy samples using neutral lysis techniques with automated scoring performed on a

**Table 3** Summary of assessments and interventions

| Time point | <30 days prior to entry | Week 1 | Week 4 | Week 8 | Week 12 | Week 16 | Week 20 | Week 24 | Week 28 | Week 32 (end of trial) |
|---|---|---|---|---|---|---|---|---|---|---|
| Enrolment | | | | | | | | | | |
| Informed consent | A+B | | | | | | | | | |
| Baseline history | A+B | | | | | | | | | |
| Eligibility screen and pregnancy test | | A+B | | | | A+B | | | | |
| Treatment allocation | | A+B | | | | | | | | |
| Conmeds | A+B | A+B | A | A | A+B | A+B | B | B | A+B | A+B |
| Interventions | | | | | | | | | | |
| Start tamoxifen (12 weeks) | | B | | | | A | | | | |
| Start anastrozole (12 weeks) | | A | | | | B | | | | |
| Goserelin implant injection | | A | A | A | | B | B | B | | |
| Assessments | | | | | | | | | | |
| Quality of life assessment | | A+B | | | A+B | | | | A+B | |
| Core breast biopsies ×3 | | A+B | | | A+B | | | | A+B | |
| Blood samples for oestrogen and metabolite levels | | A+B | | | A+B | | | | A+B | |
| Blood sample for whole blood | | A+B | | | A+B | | | | A+B | |
| Urine samples for oestrogen and metabolite levels | | A+B | | | A+B | | | | A+B | |
| Adverse event assessment | A+B | A+B | | | A+B | A+B | | | A+B | A+B |

A corresponds to treatment arm A of the study (goserelin and anastrozole followed by tamoxifen. B corresponds to treatment arm B, tamixofen followed by goserelin and anastrozole.

high content screening system. FFPE breast core biopsy blocks are created at the same time points as fresh-frozen samples. Formation of γH2AX in response to DNA DSBs provides the basis for a sensitive immunohistochemical assay of DNA damage in human biopsies and has been optimised in FFPE breast samples with simultaneous staining for 53BP1, which localises to DNA strand breaks, with foci numbers proportional to DNA damage levels. Based on our hypothesis, the anticipated decrease in measurable DNA damage in response to oestrogen suppression would support the use of this strategy as a chemoprevention option in BRCA1 mutation carriers.

Further FFPE sections will be analysed for Ki67, RANK and RANKL expression to assess the effect of treatment on these markers of proliferation (Ki67) and progesterone signalling (RANK/RANKL). As the LHRHa used to suppress oestrogen will also result in progesterone suppression, it is expected that RANK and RANKL expression will also decrease. Given that inhibition of RANK has been shown to reduce breast cell proliferation in BRCA1 mutation carriers, it is possible that LHRHa treatment might be an alternative way of downregulating this pathway and potentially decreasing the development of highly proliferative cells with grossly aberrant DNA repair observed in BRCA1 precancerous tissues with high levels of RANK/RANKL expression.[15]

Oestrogen metabolite levels in serum and urine will be measured using UPLC-MS/MS and have been optimised on healthy volunteer samples with permission from Queen's University Ethics Committee.

### Patient and public involvement

BRCA1 and BRCA2 mutation carriers were involved throughout the design process via the Northern Ireland BRCA mutation carrier support group, BRCA Link NI. The study design was finalised with input from a BRCA patient advocate focus group, where the consensus was that inclusion of an active control treatment was more

**Table 4** Amendment chronology

| Amendment number | Summary of amendment | Documents impacted (and version) | Date sent for classification | Date classified | Date sent to MHRA | Date of MHRA approval | Date sent to ethics | Date of ethics approval | Date sent to R and D | Date of R and D approval | Effective date |
|---|---|---|---|---|---|---|---|---|---|---|---|
| 1 | Contraceptive advice amended | Protocol 1.2 dated 24 May 2016 | 4 September 2016 | Minor 16 September 2016 | NA | NA | 21 June 2016 | 21 June 2016 | 4 September 2016 | NA | NA |
| 2 | Changes in the priority of existing objectives and endpoints | Protocol 2.0 dated 30 August 2016 PIS ICF 1.1 dated 19 April 2016 | 4 September 2016 | Substantial 16 September 2016 | NA | NA | 23 September 2016 | 27 September 2016 | 11 October 2016 | 19 January 2017 | NA |
| 3 | Type of blood amended and volume | Protocol 2.1 dated 9 January 2017 | 9 January 2017 | Substantial 30 January 2017 | NA | NA | 7 February 2017 | 1 March 2017 | 1 March 2017 | 14 March 2017 | 14 March 2017 |
| Minor 20 January 2017 | Addition of Keith Lowry and Lesley McFaul | NA | NA | NA | NA | NA | NA | NA | 20 January 2017 | 13 February 2017 | 13 February 2017 |
| 4 | Sample collection change | Protocol 3.0 | 4 April 2017 | Substantial 8 May 2017 | NA | NA | 8 May 2017 | 2 June 2017 | 10 May 2017 | 14 June 2017 | 15 June 2017 |
| 5 | Letter to potential participants, modify exclusion criteria | Protocol 4.0 dated 11 September 2017 | | | 29 September 2017 | 10 October 2017 | 10 October 2017 | 9 November 2017 | 16 November 2017 | | |
| Letter 1.0 dated 11 September 2017 | 11 September 2017 | Substantial 21 September 2017 | | | | | | | | | |

ICF, Informed Consent Form; MHRA, Medicines and Healthcare products Regulatory Agency; NA, not applicable; R and D, Research and Development.

acceptable than placebo. Women within the group had experienced tamoxifen therapy as part of cancer treatment and deemed potential side effects acceptable in the context of risk reduction. The BRCA mutation carrier cohort in Northern Ireland is overwhelmingly enthusiastic about chemoprevention clinical trials, resulting in a highly motivated patient group keen to contribute to future solutions to BRCA-related breast cancer risk. In keeping with this, the trial management group (TMG) for this study includes a BRCA1 mutation carrier.

## DATA COLLECTION AND MANAGEMENT
The Investigator and study site staff will ensure that data collected on each subject are recorded on the CRF as accurately and completely as possible. Data management processes will be fully defined within the data management plan for the study, held within the trial master file (TMF).

## TRIAL MANAGEMENT
### Trial Management Group
The TMG will consist of an independent chair, the chief investigator, coinvestigators and clinical research nurse, as well as other relevant committee members, as defined within the TMG charter. The TMG will provide overall supervision of the trial on behalf of the trial funder and sponsors, and ensure that the trial is conducted according to Good Clinical Practice guidelines. The TMG will focus on progress of the trial, adherence to trial protocol and patient safety. The TMG will report to the BHSCT Clinical Trial of an Investigational Medical Product (CTIMP) Oversight Committee.

Frequency of meetings will be defined within the TMG charter but as a minimum will meet at least quarterly.

### Trial Steering Committee
The BHSCT CTIMP Oversight Committee, which meets quarterly, acts as the Trial Steering Committee for chemoprevention in BRCA1 mutation carriers (CIBRAC).

### Independent data monitoring committee
After review by the sponsor, it was not considered that an independent data monitoring committee was required for this trial due to its low-risk nature, in that the IMPs are not novel agents and are licensed agents with a well-defined safety profile.

### Monitoring
The sponsor has appointed the Clinical Research Monitoring Service within the Northern Ireland Clinical Trials Unit to undertake on site monitoring. An initial monitoring visit was carried out prior to the release of the final permissions letter by the sponsor. The first interim monitoring visit will take place within 6 months of the first patient being recruited. Further monitoring visits will take place approximately every 6 months from the date of the first visit. A close-out visit will occur once the last

patient visit has been completed. A trial monitoring plan is available within the TMF.

The Standard Protocol Items: Recommendations for Interventional Trials checklist has been used in writing this report.[26]

## ETHICS AND DISSEMINATION
### Ethics
The Health and Social Care Research Ethics Committee B, as part of the Office for Research Ethics Committees Northern Ireland, awarded ethical approval for this study (REC Reference: 16/NI/0055, 21/4/16). A summary of amendments is given in table 4.

The IMPs used in this study are all licensed agents which are in common use in the treatment of breast cancer. They therefore have well-defined safety profiles available.

The women offered the opportunity to participate in this study are not patients with cancer, but rather are healthy premenopausal women who carry a pathogenic mutation in the BRCA1 gene. Although this is effectively a healthy volunteer study, it was felt it would not be appropriate to offer incentives for participation, although travelling expenses are covered for trial hospital visits.

There was extensive patient and public involvement in the study design and development, to ensure that the proposed treatments and study procedures would be acceptable to patients.

### Trial registration
CIBRAC is registered as a clinical trial of an interventional medicinal product on the European Clinical Trials Database, as EudraCT: 2016-001087-11. Trial registration data are provided in table 5.

### Dissemination
Academic dissemination of the study will be via publication in peer-reviewed journals in the relevant field (the International Committee of Medical Journal Editors criteria for authorship will be followed), and by presentation at appropriate national and international meetings. These may include meetings such as the biannual Symposium on Hereditary Breast and Ovarian Cancer in Montreal, Canada. A lay summary of the study results will be available, and all participants will be offered access to a copy of this. Furthermore, this will be shared with our partner Patient and Public Involvement (PPI) organisation BRCA Link NI and disseminated through their website and social media channels. All mass spectrometry and sequencing data generated from this study and/or the future use of study samples will be deposited in appropriate publicly accessible repositories, with the associated anonymised metadata.

### Author affiliations
[1]Centre for Cancer Research and Cell Biology, Queen's University Belfast, Belfast, UK
[2]Northern Ireland Cancer Trials Network, Belfast City Hospital, Belfast, UK
[3]BRCA Link NI, Ballynahinch, UK

**Table 5** Trial registration data

| Data category | Information |
| --- | --- |
| Primary registry and trial registration number | EudraCT: 2016-001087-11 |
| Date of registration in primary registry | 9 March 2016 |
| Sources of monetary or material support | Supported by a Cancer Research UK Clinical Research Training Fellowship |
| Sponsor | Belfast Health and Social Care Trust |
| Contact for public queries | AMC (a.campbell@qub.ac.uk) |
| Contact for scientific queries | SAM (s.mcintosh@qub.ac.uk), KIS (k.savage@qub.ac.uk) |
| Public title | CIBRAC: chemoprevention in BRCA1 mutation carriers |
| Scientific title | Chemoprevention in BRCA1 mutation carriers (CIBRAC): an open allocation crossover trial assessing mechanisms of chemoprevention of goserelin and anastrozole versus tamoxifen |
| Countries of recruitment | UK (Northern Ireland) |
| Interventions | Tamoxifen 20 mg daily (oral), goserelin 3.6 mg every 28 days (subcutaneously) plus anastrozole 1 mg daily (oral) |
| Key inclusion criteria | Ages eligible for study ≥18 years. Sex eligible for study: female. Known pathogenic germline BRCA1 mutation. Premenopausal. No previous breast or ovarian risk-reducing surgery. |
| Key exclusion criteria | Personal history of breast or ovarian carcinoma. Previous risk-reducing breast or ovarian surgery. Postmenopausal status. |
| Study type | Interventional. Allocation: alternate treatment allocation, non-randomised, non-blinded. Primary purpose: prevention. Phase II. |
| Date of first enrolment | May 2017 |
| Recruitment status | Recruiting |
| Primary outcome | Establish the acceptability of trial treatments as a chemopreventive strategy in BRCA1 mutation carriers, as measured by the number of patients entering the trial and compliance with treatment. |
| Key secondary outcomes | To establish tolerability of trial treatments and procedures. |

[4]School of Pharmacy, Queen's University Belfast, Belfast, UK
[5]Institute for Global Food Security, Advanced ASSET Centre, School of Biological Sciences, Queen's University Belfast, Belfast, UK

**Acknowledgements** We are grateful for the contribution of our patient advocates in developing the study, particularly Hazel Carson of BRCA Link NI.

**Contributors** SAM and KIS generated the study idea and hypothesis. AMC was responsible for writing the protocol, supervised by SAM and KIS. MM, RG, RB and HC contributed to protocol development, writing and revision. EW, CE and DPH contributed to experimental design and validation. DPH also contributed to protocol development. SAM is the chief investigator of the study and has overall responsibility for its conduct. All authors have seen and approved the final manuscript.

**Funding** This work is supported by Cancer Research UK as part of a Clinical Research Training Fellowship award.

**Disclaimer** The funding body has no input into the design of the study, collection, analysis or interpretation of data, nor into the writing of the manuscript.

**Competing interests** None declared.

**Patient consent for publication** Not required.

**Ethics approval** Health and Social Care Research Ethics Committee B, Office for Research Ethics Committees Northern Ireland.

**Provenance and peer review** Not commissioned; externally peer reviewed.

**Author note** Trial sponsor: Belfast Health and Social Care Trust (BHSCT) Sponsorship. Contact details: Alison Murphy, Belfast Health and Social Care Trust Research & Development Office, Room 2010, 2nd Floor, King Edward Building, Royal Hospitals Site, Grosvenor Road, Belfast, BT12 6BA. Role of sponsor: The study sponsor undertook peer review of the protocol. However, the sponsor had no role in study design, nor will they be involved in data collection, analysis, interpretation of data, report writing or decisions in respect of publication.

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
