## [Reviewer comments · BMJ Open]

ARTICLE DETAILS

TITLE (PROVISIONAL)	Chemoprevention in BRCA1 mutation carriers (CIBRAC): Protocol for an open allocation crossover feasibility trial assessing mechanisms of chemoprevention with Goserelin and Anastrozole versus Tamoxifen and acceptability of treatment.
AUTHORS	Campbell, Aideen; Morris, Melanie; Gallagher, Rebecca; Boyd, Ruth; Carson, Hazel; Harkin, D. Paul; Wielogorska, Ewa; Elliott, Christopher; Savage, Kienan; McIntosh, Stuart

VERSION 1 – REVIEW

REVIEWER	Steven Narod University of Toronto, Canada
REVIEW RETURNED	02-Apr-2018

GENERAL COMMENTS	consider the up to date literature that suggests that oophorectomy does not impact on breast cancer risk in BRCA1 carriers from two recent cohort studies consider that DNA damage is not a validated biomarker for cancer risk in carriers. this is highly speculative consider that a positive result from this study will not be sufficient in pursuing a large scale randomized trial of chemoprevention and there is no assurance patients outside of Ireland will be willing to enter such a trial. this is a major undertaking and requires international collaboration. there is no evidence yet for a buy in from others. I would not endorse my patients to enter this trial given that there are competing trials will take 2000 carriers randomized and this is challenging. currently most study centers are initiating trials with denosumab and other agents.
---

REVIEWER	Piero Sismondi Professor Emeritus of Gynaecological Oncology The University of Torino School of Medicine
REVIEW RETURNED	09-Aug-2018

GENERAL COMMENTS	Even if it seem counteintuitive to use SERMS or AIs to prevent Estrogen Receptor Negative Breast Tumors, it is widely accepted that prophylactic salpingo oophorectomy to prevent ovarian tumors in BRCA carriers is associated with a reduction of Breast Cancer Risk. Moreover the Author's preliminary laboratory data show that oestrogens and oestrogens metabolites play a role in the DNA double strand breaks in Estrogen Negative breast cells. This suggests that Oestrogen activity suppression could be effective in reducing breast cancer risk in BRCA mutation carriers as well. The study seems a sound attempt at evaluating this hypothesis.
---

	The dosage of Tamoxifen in the manuscript is erroneously stated at 1 mg whereas in the abstract is correctly set at 20 mg. This must be amended. My opinion is that the protocol is worth publishing as it is.
--	---

VERSION 1 – AUTHOR RESPONSE

Reviewer number 1:

We thank Professor Narod for his comments. We agree that there is literature suggesting that oophorectomy does not impact on breast cancer risk in BRCA1 mutation carriers, and have cited the two recent studies to which he refers (Kotsopoulos et al JNCI 2017 and Heemskirk-Gerritsen et al JNCI 2015). These studies are discussed, together with their potential limitations and conflicting data from other human and mouse studies (Introduction, paragraph 1, page 4). However, as stated, we feel that this evidence is currently inconclusive and thus further investigation is justified.

We also agree that DNA damage is not a validated biomarker for breast cancer risk. However, genomic instability is a well-characterised early hallmark of BRCA1-associated breast cancer, and we have revised the manuscript to discuss this (Introduction, paragraph 6, pages 4/5).

Finally, we would agree with Professor Narod that many groups are currently investigating progesterone and RANK/RANK-L signalling in this context, and we have revised the final paragraph of the Introduction to discuss this in more detail. However, very recently published data suggests that oestrogen rather than progesterone signalling may be the predominant driver of tumourigenesis (van de Ven et al , J Pathol 2018), suggesting that at the very least this pathway bears further investigation as in our feasibility study. We agree with Professor Narod that evidence from this study alone will not be sufficient to support an international multicentre collaborative study, and we have acknowledged this as he suggests in the final paragraph of the Introduction (page 5).

Reviewer number 2:

We have corrected the dose of tamoxifen in the manuscript from 1mg to 20mg. We apologise for this oversight.